

# The 2019 Raikoke volcanic eruption part 2: Particle phase dispersion and concurrent wildfire smoke emissions

Martin Osborne[1,2], Johannes de Leeuw[3], Claire Witham[1], Anja Schmidt[3,4], Frances Beckett[1], Nina Kristiansen[1], Joelle Buxmann[1], Cameron Saint[1], Ellsworth J. Welton[5], Javier Fochesatto[5,6], Ana R Gomes[7], Ulrich Bundke[7], Andreas Petzold[7], Franco Marenco[1,8], and Jim Haywood[1,2]

[1]Met Office, FitzRoy Road,Exeter, Devon, EX1 3PB, United Kingdom
[2]University of Exeter, Laver Building, North Park Road, Exeter, Devon, EX4 4QE, United Kingdom
[3]University of Cambridge, Department of Chemistry, Lensfield Road, Cambridge, CB2 1EW, United Kingdom
[4]University of Cambridge, Department of Geography, 20 Downing Place, Cambridge CB2 1BY, United Kingdom
[5]NASA Goddard Space Flight Center, Greenbelt, Maryland, USA
[6] Department of Atmospheric Sciences, University of Alaska Fairbanks, Fairbanks, Alaska, 99775 USA
[7]Institute of Energy and Climate Research 8: Troposphere, Forschungszentrum Jülich, Jülich, 52425, Germany
[8]Climate and Atmosphere Research Centre (CARE-C), Cyprus Institute, Nicosia, Cyprus

**Correspondence:** Martin Osborne (martin.osborne@metoffice.gov.uk)

**Abstract.** Between 27 June and 14 July 2019 aerosol layers were observed by the United Kingdom (UK) Raman lidar network in the upper troposphere and lower stratosphere. The arrival of these aerosol layers in late June caused some concern within the London Volcanic Ash Advisory Centre (VAAC) as according to dispersion simulations the volcanic plume from the 21 June 2019 eruption of Raikoke was not expected over the UK until early July. Using dispersion simulations from the Met

Office Numerical Atmospheric-dispersion Modelling Environment (NAME), and supporting evidence from satellite and in-situ aircraft observations, we show that the early arrival of the stratospheric layers was not due to aerosols from the explosive eruption of the Raikoke volcano, but due to biomass burning smoke aerosols associated with intense forest fires in Alberta, Canada that occurred four days prior to the Raikoke eruption. We use the observations and model simulations to describe the dispersion of both the volcanic and forest fire aerosol clouds, and estimate that the initial Raikoke ash aerosol cloud contained

around 15 Tg of volcanic ash, and that the forest fires produced around 0.2 Tg of biomass burning aerosol. The operational monitoring of volcanic aerosol clouds is a vital capability in terms of aviation safety and the synergy of NAME dispersion simulations and lidar data with depolarising capabilities allowed scientists at the Met Office to interpret the various aerosol layers over the UK, and attribute the material to their sources. The use of NAME allowed the identification of the observed stratospheric layers that reached the UK on 27 June as biomass burning aerosol, characterised by a particle linear depolarisation

ratio of 9%, whereas with the lidar alone the latter could have been identified as the early arrival of a volcanic ash / sulphate mixed aerosol cloud. In the case under study, given the low concentration estimates, the exact identification of the aerosol layers would have made little substantive difference to the decision making process within the London VAAC. However, our work shows how the use of dispersion modelling together with multiple observation sources enabled us to create a more complete description of atmospheric aerosol loading.



## 1 Introduction

Explosive volcanic eruptions can inject volcanic ash and sulfur dioxide ($SO_2$) into the stratosphere that can have residence times of many months or even years (e.g. Langmann, 2014; Carn et al., 2017). The dangers and disruption presented to aviation by volcanic ash aerosol clouds have been well documented, and can be present even far from the emitting volcano when long range transport takes place (e.g. Guffanti et al., 2010; Gordeev, 2014). Of particular concern to the aviation community is that volcanic ash melts in the high temperature chambers within jet engines, coating the turbines and, in high enough concentrations, can cause the engine to stall with no guarantee of it re-starting (Miller and Casadeval, 1999). Volcanic aerosol clouds containing $SO_2$ and / or sulphuric acid droplets have the potential to damage aircraft windows and cause distress to passengers and crew (Bernard and Rose, 1990). The damage to engines and airframes resulting from an aircraft encountering volcanic aerosol clouds can cause significant costs to airlines, with a single encounter potentially costing tens of millions of Euros (Miller and Casadeval, 1999; Prata and Tupper, 2009). Another impact of volcanic ash clouds (and also of desert dust layers etc.) on aviation is the effect of these aerosol layers on visibility from a pilot's perspective (Weinzierl et al., 2012). Precautionary regulatory measures resulting from volcanic activity, while necessary for public safety, can have far reaching impacts on the entire industry. For example, the 2010 eruption of the Icelandic volcano Eyjafjallajökull caused widespread disruption to air travel across Europe for several weeks and had a significant financial impact on several large airlines (Gertisser, 2010). To mitigate these risks the International Civil Aviation Authority designates nine centres to act as Volcanic Ash Advisory Centres (VAACs), each of which is responsible for issuing warnings and information to national aviation authorities and the wider aviation community. The Met Office acts as the London VAAC, and formulates guidance products using a combination of dispersion model simulations (Jones et al., 2007; Webster et al., 2012; Dacre et al., 2015), satellite-observations (Millington et al., 2012; Francis et al., 2012), and ground-based and aircraft observations (Turnbull et al., 2012; Marenco et al., 2016; Osborne et al., 2019).

Biomass burning aerosol (BBA) can also be injected to aircraft flight altitudes (e.g. Fromm et al., 2005, 2006; Christian et al., 2019; Khaykin et al., 2020), but has received far less attention from the aviation community, as it does not have a direct potential for damaging aircraft. BBA can however cause disruption to aviation as the characteristic burning smell can be misidentified as originating from the aircraft, causing activation of emergency procedures. For example, in October 2017, 32 emergency procedure and MAYDAY calls from aircraft in UK airspace were received owing to reports of smoke within the cabin leading to emergency descents, emergency landings and emergency evacuations, which caused much passenger distress and displacement of passengers and aircraft. Studies of this event have revealed that there were unusually high concentrations of BBA in UK airspace that had been transported from fires on the Iberian Peninsula by the synoptic large scale transport associated with Hurricane Ophelia (Osborne et al., 2019).

As part of its ground based volcanic ash remote sensing capability, the Met Office operates a network of 10 single-wavelength, ground-based Raman lidars distributed across the UK. The installations also have co-located AERONET sun photometers (Adam et al., 2017; Osborne et al., 2019). Active laser remote sensing using lidars is well suited to the task of detecting aerosol and cloud layers and providing quantitative estimates of concentrations as it provides atmospheric profiles that





are highly resolved in both altitude and time. Lidar networks, e.g. NASA-MPLNET (Micro-pulse Lidar Network), EARLINET
(European Aerosol Research Lidar Network) and LALINET (Latin America Lidar Network) (Welton et al., 2001; Pappalardo
et al., 2014; Guerrero-Rascado et al., 2016; Lewis et al., 2016), can also provide coverage over a wide geographical area and
can be used to track the progress and evolution of aerosol clouds. By using lidars equipped with a Raman channel as well
as depolarisation discrimination, aerosol type identification can be attempted as well as the estimation of separate mass pro-
files for depolarising and non-depolarising aerosols (e.g. Ansmann et al., 1992; Webley et al., 2008; Tesche et al., 2009; Groß
et al., 2015a). However, aerosol identification made using these methods can sometimes be ambiguous, and it is useful to use
additional information to try and reach a more certain identification. Dispersion model simulations and subsequent source attri-
bution frequently offer a good chance of arriving at an unambiguous aerosol classification. The Met Office has developed and
maintains a world-leading atmospheric dispersion model, NAME (Jones et al., 2007). In addition to its role as an emergency
response guidance tool, the model is used for routine air quality forecasting and meteorological research activities.

In late June and early July 2019 aerosol layers were observed in the UTLS by several lidars in the UK including those in
the Met Office network (Vaughan et al., 2021). While dispersion modeling performed at the time suggested that aerosols from
the eruption of the Raikoke volcano on 21 June 2019 would reach the UK by early July 2019, the arrival of aerosol layers in
the UTLS in late June caused some surprise. The depolarisation ratio measurement of 9% in these initial layers meant that the
Raikoke eruption could not be ruled out as a possible source using this evidence alone.

This paper forms part two of a two part study. In part one, de Leeuw et al. (2020) describes the transport and chemical
evolution of the gas-phase emissions from the 2019 Raikoke eruption and provides a detailed comparison of NAME model
simulations with observations from the Tropospheric monitoring instrument (TROPOMI) on-board the European Space Agency
Sentinel 5P satellite. Here, we focus on the transport and evolution of the particle phase of the Raikoke volcanic emissions
together with the transport of particulate material from fires in Alberta, Canada. For consistency with the study of de Leeuw
et al. (2020), we use the term aerosol cloud to represent the area influenced by volcanic and BBA aerosols. We acknowledge that
this is not technically meteorologically correct as clouds are formed of droplets of condensed water vapour, but we maintain
this terminology for consistency between the two studies.

In section 2 of this paper, we briefly describe the Raikoke eruption and concurrent relevant intense forest fires, and then in
section 3 describe how the NAME model simulations were initialised, as well as a brief description of the observation data and
various instruments used in this study. In section 4 we present the results, concentrating first on the geographic distribution
and vertical structure of volcanic aerosols, then on the geographic distribution and vertical structure of BBA with additional
regional analyses over North America and over the UK. Section 5 provides some conclusions.

## 2 Raikoke eruption and Wildfires in summer 2019

Over the course of 21 and 22 June 2019, the Raikoke volcano located in the Kuril archipelago in the North West Pacific erupted
explosively, injecting ash and $SO_2$ into the upper troposphere and lower stratosphere. At approximately 1.5 +/- 0.2 Tg of $SO_2$
(Crafford and Venzke, 2019; de Leeuw et al., 2020) this emission is of similar magnitude to the eruptions of Kasatochi in


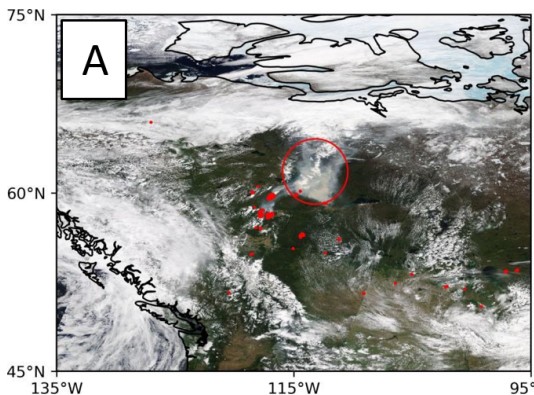
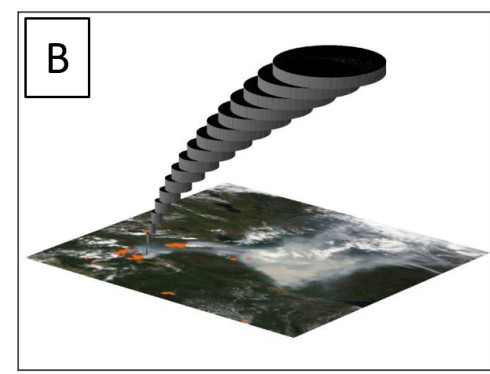

**Figure 1.** (a) MODIS Terra true colour reflectance tile for 17 June 2019 with thermal anomalies overlaid in red. Emissions from forest fires can be seen as a brownish grey cloud (circled in red). (b) Conceptual diagram of NAME source term used to initialise BBA dispersion simulations

August 2008 (Kravitz et al., 2010), Sarychev in June 2009 (Haywood et al., 2010), and Nabro in June 2011 (Clarisse et al., 2012). Satellite imagery of the plume indicates that the eruption was ash-rich in nature, estimates of the emission are derived in section 3.1.

The 2019 wildfire season in the Canadian province of Alberta was unusually severe, with more than 883,000 hectares burned across the province, compared to the five yearly average of 245,000 hectares (Short, 2019a; Jenner, 2019; Gabbert, 2019). The Mackenzie County area in the north of Alberta suffered a particularly large number of fires in May and June 2019, resulting in the evacuation of thousands of people and the destruction of property (Short, 2019b). Continued hot and dry conditions along with high winds caused intensification of the wildfires; the heat released fuelled pyrocumulus clouds, which

lofted smoke into the UTLS and allowed high-level winds to transport it long distances. On 17 June 2019, four days prior to the eruption of Raikoke, wildfires burning in Chuckegg Creek in Mackenzie County produced biomass burning aerosols and formed pyrocumulus clouds which breached the tropopause and entered the lower stratosphere. Figure 1a shows a true colour reflectance MODIS image from the 17 June 2019 overlaid with thermal anomalies associated with forest fires; an aerosol cloud can clearly be seen to have moved north-eastwards from an area with several active fires. We will show that the aerosol cloud

continued eastwards over the following days, with the lower tropospheric layers moving south over the United States and the stratospheric layers continuing eastwards to reach Europe by 26 June.



## 3 Methods

### 3.1.1 NAME dispersion model simulations

The dispersion of gas and aerosols is simulated within NAME by tracking notional 'particles' as they move within air parcels
advected by the wind vectors supplied by a driving meteorological model. Operationally the Met Office Unified Model (UM)
version 7.1 (Walters et al., 2019) provides the global driving meteorology with a three-hourly temporal resolution and approxi-
mately 10 km horizontal resolution at mid-latitudes, and on 59 model levels between the surface and 30 km altitude. Turbulence
and sub-grid scale diffusion are represented within NAME by the addition of random perturbations to the wind vectors at each
time step (Webster et al., 2012). de Leeuw et al. (2020) make a detailed quantitative investigation of the effect of this diffusion
parameter on the spatial extent and concentration of the diffused volcanic $SO_2$ cloud. For the Raikoke eruption, de Leeuw
et al. (2020) obtained better agreement between the simulated and observed $SO_2$ clouds by using a reduced parametrisation
of sub-grid scale diffusion, although the improvements were confined to the first five days after the eruption. The BBA and
ash simulations described here were run using both the default and the reduced diffusion parameter. However, the satellite
observations of the relatively low aerosol concentrations emitted by the Raikoke eruption and the Alberta forest fire were gen-
erally of poorer quality than the TROPOMI $SO_2$ observations, and we found no substantive improvement between the aerosol
simulations with the default and reduced diffusion parameter, when compared with the available observations. Therefore, only
the results using the default values for diffusion as detailed in Webster et al. (2012) are shown here.

NAME simulations are initialised by defining source terms that describe the location, vertical profile and mass emission rate
of the event producing the gas or aerosol being simulated. Where relevant, there is the option to define further characteristics
such as chemical species, size distribution and particle shape. The emitted material is then subject to chemical and/or physical
transformations and can be removed by gravitational settling, wet and dry deposition. Relevant to this study is the chemical
transformation of sulfur dioxide ($SO_2$) into particulate sulfate aerosol ($SO_4^{2-}$). This process is represented within NAME
by gaseous and aqueous phase chemistry schemes. The species involved in these reactions are simulated explicitly within
NAME and are initialised at the start of a model run using values taken from background fields provided by the Met Office
Unified Model coupled to the United Kingdom Chemistry and Aerosol model (UM-UKCA). Redington and Derwent (2002)
& Heard et al. (2012) provide full details of the equilibrium equations used in the sulfur chemistry scheme. Within the NAME
simulations BBA and volcanic ash particles are not subject to chemical processing, but are subject to relevant physical processes
as described in the following sub-sections.

Gravitational settling and particle shape and size - Particulates, including BBA and volcanic ash, are subject to sedimentation
by gravitational settling, which is expressed in NAME by a particle's overall vertical velocity. This vertical velocity is a com-
bination of wind advection, turbulent diffusion and sedimentation fall velocity. The model calculation of sedimentation fall
velocity accounts for the particle's physical properties as well as the viscosity and density of the surrounding atmosphere.
NAME includes options to use one of several "fall schemes", each of which uses different formulae to calculate the sedimenta-
tion fall velocity. Saxby et al. (2018) compared the fall velocities of non-spherical volcanic ash particles calculated by various





**Table 1.** Cumulative aerosol size distributions for BBA and Volcanic ash according to Hobbs et al. (1991); Maryon et al. ((1999); Petzold et al. (2007),used in the NAME simulations.

| Biomass burning aerosol | | Volcanic ash | |
|---|---|---|---|
| Diameter [$\mu$m] | Cumulative fraction [%] | Diameter [$\mu$m] | Cumulative fraction [%] |
| 0.01 | 0 | 0.1 | 0 |
| 0.03 | 0.004 | 0.3 | 0.001 |
| 0.1 | 0.037 | 1.0 | 0.006 |
| 0.3 | 0.75 | 3.0 | 0.056 |
| 1.0 | 0.89 | 10.0 | 0.256 |
| 3.0 | 0.99 | 30.0 | 0.956 |
| 10.0 | 1.0 | 100.0 | 1.0 |

schemes to laboratory measurements and found that the fall scheme described by Ganser (1993) was the most accurate. We have therefore adopted this scheme in our simulations. As shown in Beckett et al. (2015) the aerosol particle size distribution (PSD) used to initialise NAME aerosol simulations has a significant impact on the number of particles removed by gravitational settling, and therefore on the number that survive in the distal aerosol cloud. Without in-situ sampling, or samples collected near the fires or volcano, it is not possible to know the size distribution or particle shape of the emissions from either the fires

or the Raikoke eruption and this distribution must be assumed. For operational ash model simulations, the London VAAC uses a default PSD that is based on in-situ measurements of the 1990 Mount Redoubt ash cloud (Hobbs et al., 1991; Maryon et al., (1999), and in-lieu of measurements of the Raikoke ash, we have used this PSD in our simulations. For the BBA cloud we have used a size distribution from in-situ measurements for Canadian forest fire emissions from Petzold et al. (2007). The distribution of mass over these size distributions is shown in table 1. Saxby et al. (2018) describe how for real particles for a

given volume and density, non-spherical particles fall more slowly than spheres owing to their larger projected area, and that therefore the particle shape used when initialising NAME also has an impact on the number of particles surviving in the distal field. NAME uses a "sphericity" parameter to describe the degree of non-sphericity of a particle, taking a value between 0 and 1 (with the sphericity of a sphere being equal to 1). The sphericity parameter is the ratio of the surface area of a sphere with equivalent volume to the actual surface area of the particle. To try to best capture the long-range transport of large grains, we

have used a sphericity parameter of 0.5 for our simulations of volcanic ash. For BBA, which we expect to be nearer spherical (e.g. Taylor et al., 2020), the sphericity parameter was set to 1.

### 3.1.2 Mass emissions and vertical profiles

Volcanic ash - For volcanic ash simulations, the approach used by the London VAAC is to assume that most of the larger grains of the total mass ejected by an eruption fall out close to the volcano, leaving only a fine mass fraction (FMF) available to be



dispersed. Operationally this FMF is set somewhere between 1% and 5% of the total ejected mass, with 95% to 99% of the total

mass assumed to have fallen out close to the volcano. To estimate the amount of ash ejected during the 2019 Raikokeeruption,

we have used the relationship between plume top height and mass eruption rate presented in Mastin et al. (2009). Using a

plume top height of 15 km as reported by the Volcano Response group (VolRes - https://wiki.earthdata.nasa.gov/display/volres

- an international climate initiative from the WMO-SPARCSSiRC working group (http://www.sparc-ssirc.org)), and eruption

start and end times of 21:00 (UTC) on the 21 June to 03:00 (UTC) on the 22 June (taken from HIMAWARI-8 imagery) we

estimated a total ejected amount of ash of around 300 Tg. We use this estimate as a representative figure. We note that the

eruption was not continuous during this time, but rather progressed in several pulsed phases, each of which possibly reached

different altitudes - and as such, this is an upper estimate. Using this value gives a FMF of between 3 and 15 Tg. Again taking

the upper limit, we released a total of 15 Tg of ash over a vertical profile taken from the VolRes report (fig. 2 in red). This

vertical profile was formulated from a best estimate for the Raikoke $SO_2$ emission profile using IASI measurements. In lieu of

a specific profile for ash, we have adopted this $SO_2$ vertical profile as the best available estimate for ash. Note however that ash

is not necessarily released in the same profile as the gas emissions.

$SO_2$ - For our volcanic $SO_2$ simulations we use the source term and dispersion parameters from de Leeuw et al. (2020) that have

been optimised against TROPOMI observations. This source term uses the height profile (shown in fig. 2 in orange) referred

to as "Stratprofile" in de Leeuw et al. (2020) using a total emission of 1.57 Tg of $SO_2$. In this profile, a larger fraction of the

total mass is released into the stratosphere compared to the un-optimised profile reported in the VolRes report (and as described

above is used for the volcanic ash source term). As described in de Leeuw et al (2021), differences between the vertical profiles

do not necessarily make them inconsistent because the VolRes profile is derived closer to the time of the eruption while those

derived in de Leeuw et al. (2020) are subsequent to potential additional lofting caused by the absorption of solar radiation

(Muser et al., 2020).

Sulphate aerosols - The sulphate aerosol results presented below are the result of the conversion of the simulated $SO_2$ into

sulfate by the chemistry scheme within NAME, and no primary sulphate was emitted in the Raikoke simulation source term.

Biomass burning aerosol - The location and vertical profile for the source term used to initialise the BBA dispersion simulation

was derived from a MODIS image from the 17 June 2019 (shown in Fig. 1a). In this image a smoke cloud has already been

advected in a north easterly direction from an area of active forest fires. While there are no measurements available to show

the height of the aerosol cloud, we assume that it is a hot and buoyant aerosol cloud that has continued to rise into the lower

stratosphere as it was advected away from the source fires (Fromm et al., 2005, 2006). To represent this in NAME we have

used a series of stacked and offset cylinders. A conceptual diagram of this structure is shown in fig. 1b. An arbitrary mass of

0.1 Tg of material was released in each of the cylinders and the results later scaled to match observations (see section 4).



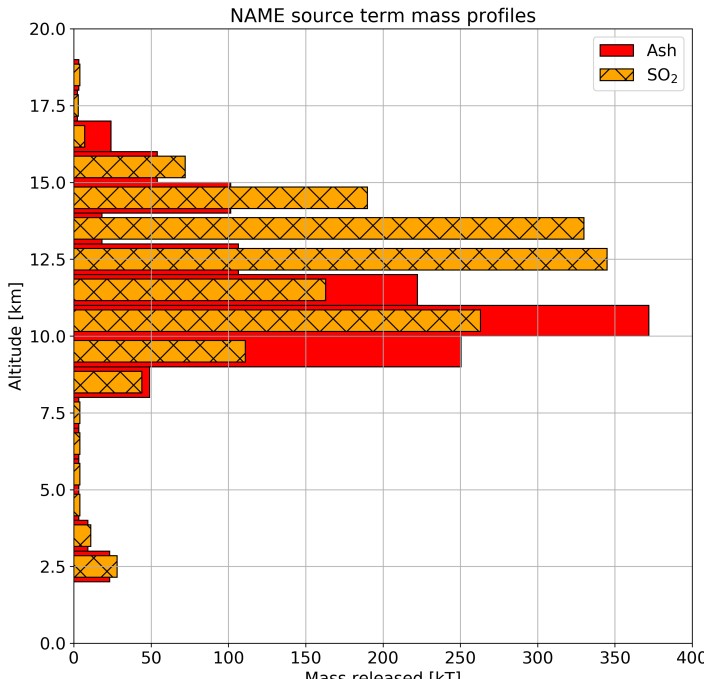

**Figure 2.** Height profiles of mass of ash and $SO_2$ released in NAME dispersion simulations of Raikoke. The ash profile (in red) was taken from the VolRes group report which used IASI satellite observations to arrive at an emission profile for $SO_2$. In lieu of a profile for ash only this profile has been adopted for the ash simulations. Plotted in orange (and hatched) is the 'StratProfile' from the part 1 paper - this is the altered profile for the release of $SO_2$ arrived at following a detailed comparison between the SO2 product from TROPOMI and NAME simulations. In this emission profile a larger fraction of the total mass of $SO_2$ is released in the stratosphere than in the original VolRes profile.

### 3.1.3 NAME output

Having initialised the simulations as described, NAME was configured to provide output files for a) column loading (two dimensional -latitude x longitude), and b) air concentration (three dimensional - latitude x longitude x altitude - with altitude measured from sea level). The data in each hourly file represents the mean of the simulated values calculated at ten-minute intervals in the preceding hour.

### 3.2 Observations

In this section we briefly describe the observations used to track the various aerosol clouds. The observations, platforms and applications in this study are summarised in Table 2.



**Table 2.** Summary of observations.

| Observation source | Products | Date | Locations used | Application in this study |
|---|---|---|---|---|
| Lidars (UK network and MPLNET) | Aerosol backscatter (at 355nm and 532nm) PDR (at 355nm and 532nm) Mass concentration | **UK lidars** 27 June to 14 July 2019 **MPLNET Fairbanks** 24 to 28 June 2019 **MPLNET Washington DC** 22 to 24 June 2019 | UK Fairbanks, Alaska Washington DC | Quantitative observation of aerosol layers. PDR provides information on aerosol type. |
| Himawari 8 | Dust RGB images | 22 to 25 June 2019 | Vicinity of Raikoke volcano / Bering sea | Information on observable horizontal area of Raikoke ash aerosol cloud. |
| OMPS | UV Aerosol index | 18 to 24 June 2019 | North Americ and Atlantic ocean | Semi quantitative information on horizontal extent of the observable BBA |
| CALIOP lidar | Aerosol backscatter @ 532 | 18 to 25 June 2019 | Vicinity of Raikoke volcano / Bering sea | Observation of vertical structure of aerosol clouds |

### 3.2.1 Surface-based remote sensing

UK lidars - The Met Office VA lidar network consists of nine fixed locations and one mobile facility (Osborne et al., 2019). The instruments emit at 355 nm and have polar and co-polar depolarisation detection channels at 355 nm and an $N_2$ Raman detection channel at 387nm. Aerosol optical properties are calculated from lidar analogue and photon-counting signals using a data analysis package developed at the Met Office which has been verified against the EARLINET Single Calculus Chain (SCC) (D'Amico et al., 2016; Mattis et al., 2016). Errors are estimated using the Monte Carlo method described in D'Amico et al. (2016). In section 4 we use the particle linear depolarisation ratios (PDRs) measured by the lidars to assist with making aerosol identifications. Each lidar is calibrated using the $\pm45^o$ procedure from Freudenthaler et al. (2009) and Freudenthaler et al. (2018). Additionally, we use the procedure described in Freudenthaler (2016) to correct for the polarisation effects of the various optical elements in the lidar. For each lidar we have also measured the angular offset between the plane of polarisation of the emitted laser beam, and that of the optical elements used to split the incoming light by polarisation state, using a method similar to that described in Alvarez et al. (2006), and any measured offset has been included in the polarisation corrections. To





**Table 3.** Aerosol specific extinction values.

| Aerosol type | Specific extinction ($\mathbf{K}_{ext}$)[m$^2$g$^{-1}$] | Comments |
|---|---|---|
| Ash | 0.6 @ 355nm and 532nm | Ansmann et al. (2011) |
| | | Marenco et al. (2011) |
| Sulfate | 4.95@355nm | Mie scattering theory |
| | 4.2@532nm | PSD from Thomason and Peter (2006) |
| | | R.I. = 1.42 - 0.0i @ 355 nm |
| | | R.I. = 1.44 - 0.0i @ 532 nm |
| | | (Palmer and Williams, 1975) |
| | | Rho = 1.77 gcm$^{-3}$ (Thomason and Peter, 2006) |
| BBA | 9.8 @ 355nm | Mie scattering theory |
| | 5.6 @ 532nm | PSD from Petzold et al. (2007) |
| | | R.I. = 1.5 – 0.08i @355nm |
| | | R.I. = 1.45-0.07i @ 532 nm |
| | | (Bluvshtein et al., 2017) |
| | | Rho = 1.9 gcm$^{-3}$ (Bluvshtein et al., 2017) |

confirm the robustness of the calibration, the volume depolarisation ratio is measured in pristine air and compared against a
theoretical value calculated as in Behrendt and Nakamura (2002) and checked to see that it is within 1% of expected values.

NASA-MPLNET lidars – The NASA Micro Pulse Lidar (MPLNET) network currently has 17 active sites around the globe,
each with a depolarisation MPL and a co-located Cimel sun-photometer (Welton et al., 2001; Welton and Campbell, 2002;
Welton et al., 2018). The instruments operate at 532 nm and also have co-polar and cross-polar depolarisation detection. To
study the vertical structure of the Raikoke and BBA clouds we make use of data from the MPLNET sites at Fairbanks in Alaska
(64.86N, 147.85E) and the Goddard Space Flight Center in Washington DC (38.99N,76.84E). We make use of the products
from the new version 3 processing system which provides normalized relative backscatter (NRB), volume depolarisation ratio
(VDR) as well as particle linear depolarisation ratio. The data is available for download from https://mplnet.gsfc.nasa.gov/ (last
accessed 16/7/2020).

Using data from UK and MPLNET lidars we have derived aerosol mass concentration estimates using the aerosol specific
extinction values ($\mathbf{K}_{ext}$) listed in table 3 to convert extinction profiles to mass estimates. For sulfate and BBA aerosols we
have used literature values for refractive index and stratospheric size distributions in Mie scattering calculations, along with
literature values for density, to calculate a representative $\mathbf{K}_{ext}$. In the case of irregularly shaped volcanic ash particles, for
which Mie calculations are unsuitable, we have used a literature value. We do not take into account factors such as hygroscopic
growth or particle cohesion etc. that will have an effect on scattering efficiency and these values are representative only. A more



complex treatment of K$_{ext}$ is beyond the scope of this study. Errors in the mass estimates have been calculated by extending the Monte Carlo technique used to take into account the statistical errors in the raw lidar signals to include errors in the assumed depolarisation ratios and K$_{ext}$. In total, the error in the mass estimates is of the order of ±50%.

Dust RGB images - In section 4 we present dust RGB images made using infrared bands B11 (8.6 $\mu$m), B13 (10.4 $\mu$m) and
B15 (12.4 $\mu$m) from the advanced HIMAWARI imager carried by the HIMAWARI 8 satellite and compare these with the NAME model simulations for volcanic ash (Francis et al., 2012). The RGB images were created by assigning red to the difference between B15 and B13, green to the difference between B13 and B11, and blue to B13. In this scheme, aerosol clouds containing volcanic ash should appear bright red, or bright yellow if there is also significant SO$_2$ (Francis et al., 2012). While not a quantitative measure, the RGB imagery is useful for assessing the horizontal spatial distribution of volcanic ash
clouds during the initial dispersion.

UV Aerosol index – The Ozone Mapping Profiler Suite (OMPS) carried in the Suomi NPP satellite measures backscattered ultraviolet radiance. The UV aerosol index (AI) is derived from radiances at 340 nm and 378.5 nm, with a temporal resolution of 24 hours (Torres et al., 1998, 2018). A positive AI value indicates the presence of UV absorbing aerosols such as dust (Christopher et al., 2008), volcanic ash (Carn and Krotkov, 2016) or smoke (Torres et al., 2020). The parameter is unit-
less and takes a value between 0.0 and 5.0, with 5.0 indicating heavy aerosol loads. The AI is available from https://snpp-omps.gesdisc.eosdis.nasa.govdataSNPP_OMPS_Level2 (date last accessed 21/06/2020). The product is semi-quantitative, and is used in section 4 to assess the horizontal spatial distribution of the BBA cloud from the Alberta fires.

CALIOP backscatter and aerosol vertical feature mask – The CALIOP lidar on board the CALIPSO satellite measures elastic backscatter and depolarisation at 532 and 1064 nm (Winker et al., 2009). The level 2 data downloaded, available from
http:www-calipso.larc.nasa.gov (last accessed 10/07/2020) also includes a vertical feature mask (VFM), which identifies aerosol layers and assigns an aerosol subtype - for example dust, volcanic ash or elevated smoke (Omar et al., 2009). In section 4 we use CALIPSO data to assist with tracking both the volcanic and BBA aerosol clouds as they travel to the UK.

### 3.2.2   IAGOS aircraft-based observations

The European research infrastructure In-service Aircraft for a Global Observing System (IAGOS; see www.iagos.org for
details) performs routine measurements of basic atmospheric state variables, atmospheric chemistry and aerosols using their standard payload on a fleet of passenger aircraft equipped with IAGOS-CORE instruments (Petzold et al., 2015). In addition to the standard equipment, a single Lufthansa Airbus 340-600 carries the more sophisticated IAGOS-CARIBIC (Civil Aircraft for the Regular Investigation of the atmosphere Based on an Instrument Container) instrumentation (Brenninkmeijer et al., 2007). During the time-period of our investigation, the IAGOS-CARIBIC aircraft made several chance penetrations of the
BBA cloud originating from Alberta, making measurements with the IAGOS aerosol instrument (Bundke et al., 2015), that contains an optical particle counter (Grimm Model 1.129 Sky) and a set of condensation particle counters (GRIMM Model





**Figure 3.** Column one: Himawari 8 Dust RGB for dates shown, columns two and three: NAME simulated column loadings for ash and sulfate clouds [gm$^{-2}$]. Both the HIMAWARI data and the NAME simulation data are for 15:00 UTC on the date shown. The red and blue triangles in the upper right panel show the locations of the Raikoke volcano and the Fairbanks MPLNET lidar respectively. The blue line in the central column shows the CALIPSO track at that time.

5.411. The optical particle counter used here measures particle concentration (as number of particles per cm3) for particle diameters between 0.25 $\mu$m and 2.5 $\mu$m.



## 4   Results and discussion

In this section we use the NAME simulations and the various observational data to chart the course of the aerosol clouds produced by the Raikoke eruption and the Alberta forest fires as they were transported to the UK, before using the simulations to assist in interpreting the UK lidar data. We first look at the initial five to six days subsequent to the emission of the ash and BBA, and then use the NAME simulations to track the transport to the UK.

### 4.1   Volcanic aerosols: transport

The left-hand column of fig. 3 shows the HIMAWARI 8 dust RGB images for the first four days after the 21 June Raikoke eruption. The central column shows the NAME simulated column loadings for volcanic ash, and the left-hand column shows the NAME simulated column loadings for sulfate aerosol. In the simulations both the ash and sulfate aerosol clouds initially move east from the volcano while also dispersing to the north and south. The leading edges of the aerosol clouds then begin to be entrained by a low pressure cyclonic system centred at around $55^{o}$N and $175^{o}$W (seen as a green and brownish cloud

in the RGB image) and turn northwards over the Aleutian Islands. The eastern ends of both the ash and sulfate clouds split into two distinct structures. Figure 3 shows that the model, the aerosol clouds continue to disperse and reach Russia by the 24 June and Alaska by the 25 June. As expected, the sulfate aerosol concentration is initially relatively low, and increases as the oxidation of $SO_2$ progresses. The RGB images reflect the general structure of the most intense parts of the model-simulated aerosol clouds and show the aerosol cloud moving east and being entrained in the warm conveyor of the cyclonic system before

encircling it and dispersing to the east and west. Figure 3 shows that on the 22 June the northern part of the aerosol cloud is bright yellow - indicating the presence of SO2 as well as ash, whereas the south-westerly part is bright red, indicating that the infrared emission in this region was dominated by ash (Francis et al., 2012). Several of the model structures are absent however, and this may in part be due to an overlying cloud obscuring the aerosol cloud from the sensor. For example, on the 23 June the most southerly structures in the model aerosol clouds are co-located with an area of cloud in the RGB images, which

appear as green and brownish colours. After 25 June the aerosol cloud can no longer be discerned in the RGB product.

### 4.2   Volcanic aerosols: vertical structure

Figure 4 shows the model ash and sulfate aerosol cloud air concentrations along the CALIPSO tracks shown in fig. 3, alongside the CALIPSO attenuated backscatter at 532nm and the VFM. On 22 June the CALIPSO track passes over the separated part of the aerosol cloud that is shown on both the RGB observation and the simulations. The northern part is shown by CALIPSO

at around 17 km, and has been classified at volcanic ash in the CALIPSO VFM (9 = light grey), and the southern part is found to be lower at under 5 km. This lower part is identified in the VFM as mineral dust (2 = yellow). Kim et al. (2018) explain that only aerosols in the stratosphere can be identified as volcanic ash or sulphate by the VFM. Below the tropopause volcanic ash will often be classified as dust or polluted dust and volcanic sulfate will often be classified as elevated smoke. This could explain the way these two plumes are captured in the CALIPSO VFM. Note that as seen in the dust RGB, both the Northern

(stratospheric) plume and the Southern (tropospheric) are weak enough as not being identified in the dust RGB at the location





of the CALIPSO overpass; however, both plumes become more intense further East, and the difference in colour in the dust RGB is indicative of a strong signal from $SO_2$ in the Northern aerosol cloud.

On 23 June the effect of the cyclonic system is evident with the two vertical structures at around $52^oN$ and $56^oN$ indicating the walls of this central pressure low. The tops of these structures are visible in the CALIPSO attenuated backscatter data. The
structure at around $43^oN$ below 5 km and identified as dust in the VFM is not visible in the RGB image, possibly because it is occluded by the cloud at around 10 km. In areas where the model places ash in the troposphere, the VFM has identified this as dust or polluted dust (2 = yellow and 5 = brown respectively), and sulfate has sometimes been identified as elevated smoke (6 = black). Almost all of the stratospheric layers identified in the VFM have been given the "volcanic ash" classification (9 = light grey), with very few "sulphate/other" classifications. According to Kim et al. (2018) the distinction between the two
classifications for layers that are optically thicker than 0.001 sr$^{-1}$ (vertically integrated attenuated backscatter) is based on the PDR: if it is above 15% the layer will be classified as "volcanic ash". Depending on the relative concentrations, a mixture of sulfate aerosols (PDR $\approx$ 1% (e.g. Ansmann et al., 2012)) and strongly depolarising volcanic ash (PDR $\approx$ 35% (e.g. Ansmann et al., 2011)) may well result in an overall PDR of more than 15% (Groß et al., 2013) - and the designation of these layers as ash in the VFM is not inconsistent with an ash-rich mix of ash and sulfate aerosols.

After having analysed the volcanic plume dispersion over the ocean using CALIPSO, we can keep following the ash and the sulphate aerosol over the American continent from ground-based remote sensing instruments. Figure 5a shows the attenuated backscatter data from the NASA-MPLNET lidar in Fairbanks, Alaska (64.86$^oN$, 147.85$^oE$) from the 24 to the 28 June 2019. Late on 24 June a banked structure (red outline 1) can be seen starting at 10 km and descending to around 5 km by 26 June. While this structure is mostly dominated by meteorological cloud, it appears to contain some aerosol, noticeable in the areas
with no cloud. The data does not allow the retrieval of a PDR or mass concentration in this structure. Above this structure, (red outline 2) a vertically thin aerosol layer can be seen at around 12 km, persisting from early on 25 June until around midday on 27 June. The whole day mean PDR within this layer on 26 June is 25%, consistent with a mixture of ash and sulfate particles (Ansmann et al., 2011; Groß et al., 2013). Using assumed values for PDR for ash (35%) and sulfate (0.01%) together with an ash-like specific extinction of 0.6m$^{-2}$g$^{-1}$ (Ansmann et al., 2011; Marenco et al., 2011), and a Mie calculated sulfate
specific extinction of 4.2m$^{-2}$g$^{-1}$ to convert the particle extinction measurements into separate mass concentration estimates for ash and sulfate, the mean estimates within this layer are 250 $\mu$gm$^{-3}$ and 10 $\mu$gm$^{-3}$ for ash and sulfate respectively. Given the uncertainty of the assumptions for PDR and specific extinction, the error on these mass estimates is on the order of 50% (Ansmann et al., 2011; Osborne et al., 2019). A second banked structure (red outline 3) starts at 11 km at midnight on the 26 June, and descends to the boundary layer top by mid-afternoon on the 27 June. While mostly obscured by low tropospheric
clouds at around 2.5 km, there are indications of aerosol layers observable in this structure. Again, the sparse data does not allow the retrieval of either a PDR or mass concentration within this structure.

Panels B & C in fig. 5 show the NAME simulated air concentrations for ash and sulphate respectively. The outlined regions from panel A are also marked in panel B with dashed lines for reference. The structure of the two simulated aerosol clouds are similar to each other, with an initial sloping structure starting at around 10 km on the 25 June, descending to 4 km by midday
on the 27 June, and a geometrically thin layer at 11 to 13km. Such sloping lidar features are frequently observed owing to

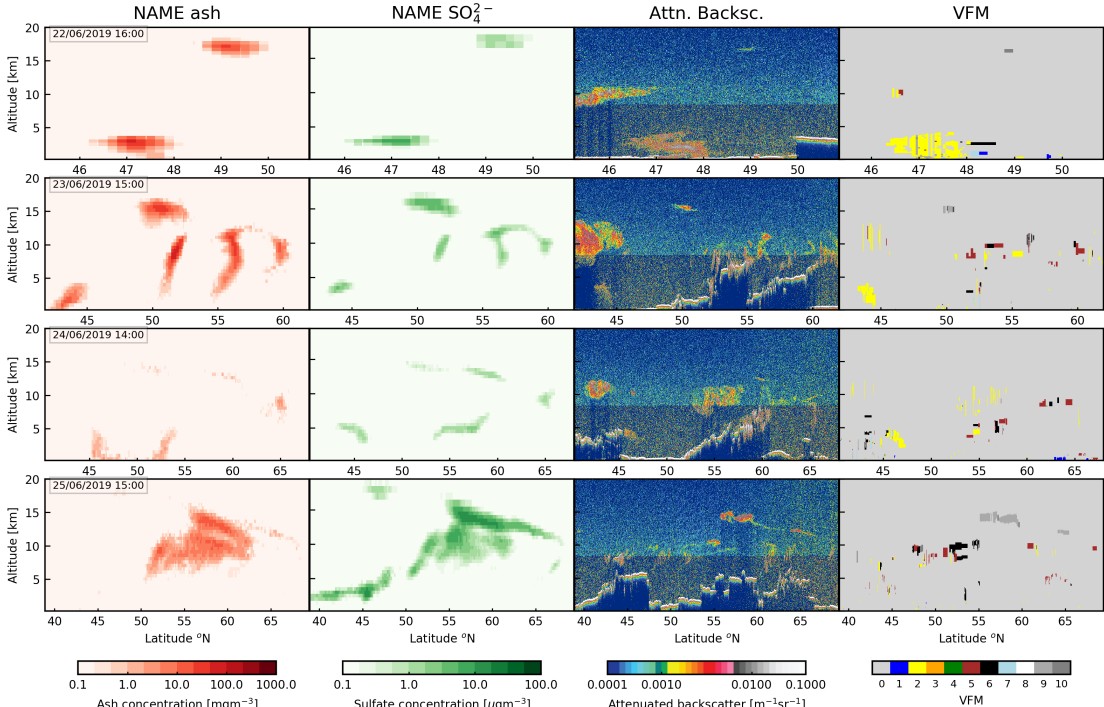

**Figure 4.** Columns one and two: Vertical distribution of simulated ash and sulfate aerosol clouds along the CALIPSO tracks shown in fig. 3, Column three: CALIPSO Total Attenuated Backscatter at 532nm, and column four CALIPSO VFM. The model data is taken from the one hour output file closest to the time of the CALIPSO overpass. Key to VFM colour bar: 0 = not applicable, 1 = marine, 2 = dust, 3 = polluted continental / smoke, 4 = clean continental, 5 = polluted dust, 6 = elevated smoke, 7 = dusty marine, 8 = polar stratospheric cloud aerosol, 9 = volcanic ash, 10 = sulfate / other.

vertical shear in the atmosphere with faster transport at higher altitudes (e.g. Dacre et al., 2015). Both of these structures are somewhat representative of the outlined structures 1 and 2 in the lidar data. However, there is a large structure between 5 km and 10 km in the simulations that is not present in the lidar observations. Attempts at modifying the vertical structure of the ash source term by changing the mass released at each altitude could not remove this structure. The maximum concentrations

in the higher layers between the times indicated by the vertical red lines is 400 $\mu$gm$^{-3}$ and 25 $\mu$gm$^{-3}$ for the ash and sulfate simulations respectively. While these values are higher than those estimated from the lidar data, they are of the same order of magnitude (and their ratio is more or less the same), giving some indication that the mass of ash allowed to remain in the distal field, and the amount of sulfate produced in the simulations by the chemistry scheme are somewhat representative of the real values.





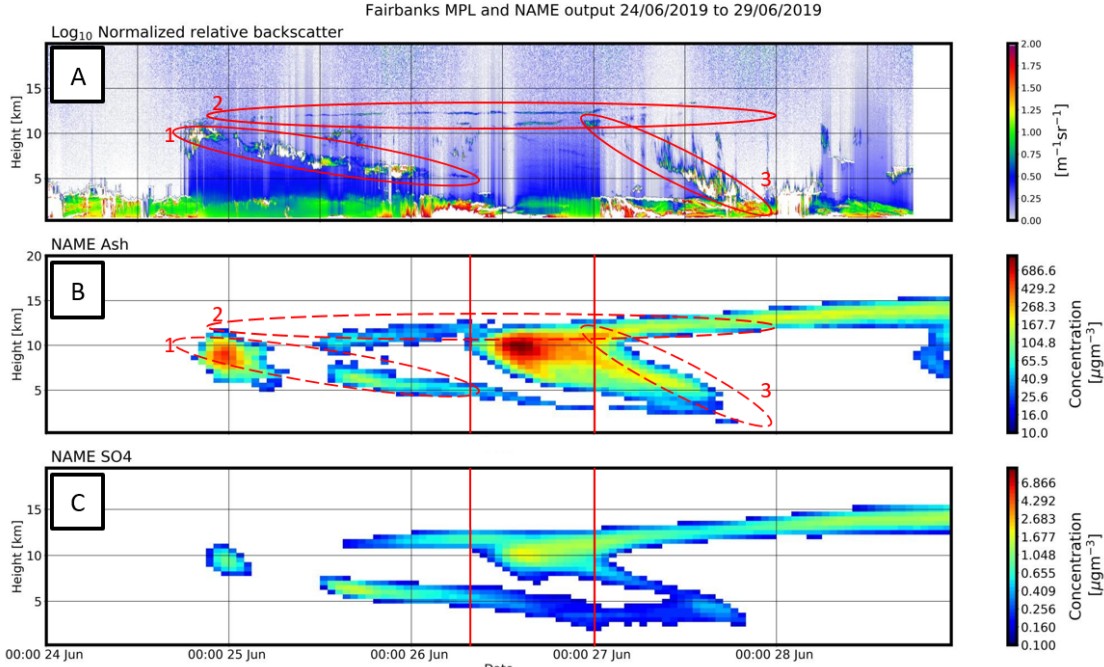

**Figure 5.** Left hand column: Daily composites of OMPS UV Aerosol index. Right hand column: Simulated column loadings for BBA (black - mgm-2) and UM dust column loadings (yellow -mgm-2) for 12:00 UTC on the date shown. The source location for simulated BBA particles is shown in the upper most right hand panel as a red triangle, and the location of the Washington MPL lidar station is shown as a blue triangle). In the right hand panels a selection of CALIPSO overpasses are marked in magenta, and four CARIBIC flight tracks are marked in cyan.

**Table 4.** Comparison between OMPS AI and NAME BBA simulation

| Date | Grid boxes in simulated aerosol cloud | Of those boxes, how many contain aerosol index observations | Percentage agreement |
|---|---|---|---|
| 17/06/2010 | 245 | 240 | 98 |
| 18/06/2010 | 2094 | 2017 | 96 |
| 19/06/2019 | 3396 | 3278 | 97 |
| 20/06/2019 | 4760 | 4363 | 92 |
| 21/06/2019 | 6230 | 4557 | 73 |
| 22/06/2019 | 9160 | 5665 | 62 |
| 23/06/2019 | 14424 | 8584 | 60 |
| 24/06/2019 | 15749 | 10612 | 67 |
| 25/06/2019 | 14444 | 9835 | 68 |





### 4.3 Biomass burning aerosol: transport

Figure 6 shows the OMPS AI, together with the NAME simulated BBA column loadings for 18 June to the 24 June. In both the simulations and the AI the BBA cloud initially moves fairly uniformly north east towards Hudson Bay for around 24 hours, before the leading edge is caught in an anticyclonic system and part of it begins to turn south before looping back and turning north and then east once again by 19 June. On 20 June the easterly edge of the aerosol cloud has crossed Hudson Bay and, in both the measurements and the simulation, the BBA aerosol cloud is also starting to extend westwards towards the Pacific coast. By 21 June the north eastern edge of the aerosol cloud has reached the Labrador coast, while the southern edge has been caught in a south easterly jet and is moving across the continental United States towards Washington DC. The western extent of the model aerosol cloud is now reaching to the Pacific coast, but this is not the case in the AI, and this part of the simulated aerosol cloud is absent from the measurements. By 22 June the leading edge of the southerly aerosol cloud has moved beyond Washington DC and out over the Atlantic where it has been entrained in a cyclonic system centred off the coast of Newfoundland and turned northwards. The more northerly part of the aerosol cloud is moving towards Greenland, and over Iceland by the 23 June. By 24 June, this more northerly part has now turned south and moved over the Atlantic towards the West of Ireland. The southerly part of the aerosol cloud has continued to be entrained by the cyclonic system and has itself moved north to Greenland. Missing entirely from the AI is the large structure that extends out to the Pacific coast. This feature is also not present in any CALIPSO profiles. This part of the simulated aerosol cloud originated almost entirely from material released at the very top of the model source, between 16 km and 18 km, and this suggests therefore that the BBA cloud did not extend above 16 km. In terms of spatial extent and horizontal structure, many of the features of the simulated BBA cloud are reflected in the AI, particularly in the first three to four days. However, the simulated aerosol cloud looks to be more diffuse than is suggested by the AI observations. This may in part be due to the detection limits of the AI, which is unlikely to be sensitive to some of the very low column loadings suggested by the model simulations. For example, as is shown in section 4.4, CALIPSO lidar soundings along the tracks marked in panels j, l and n of fig. 6 detect elevated smoke in regions without significant AI index values (see panels i, k and m of fig 6), showing that the BBA cloud can be present even where no AI index is observed. A quantitative comparison between the AI index and NAME simulations is difficult not only because the lower concentrations can be missed by the AI index, but also because the AI index is sensitive to other common aerosols (for example dust – see Met Office operational dust forecast plotted in fig. 6), and so will always detect significant amounts of background aerosol. However, to give some quantification within the bounds of the simulated BBA cloud we have re-gridded the AI onto the same latitude / longitude grid as the NAME simulations and compared the number of grid boxes within the simulated aerosol cloud that also have a positive AI index. The results of this comparison are shown in table 4. For the first four days following emission over 90% of the grid boxes within the simulated BBA cloud also contain an AI observation. This then falls to as low as 60% over the next four days.





**Figure 6.** Left hand column: Daily composites of OMPS UV Aerosol index. Right hand column: Simulated column loadings for BBA (black - mgm$^{-2}$) and UM dust column loadings (yellow - mgm$^{-2}$) for 12:00 UTC on the date shown. The source location for simulated BBA particles is shown in the upper most right hand panel as a red triangle, and the location of the Washington MPL lidar station is shown as a blue triangle). In the right hand panels a selection of CALIPSO overpasses are marked in magenta, and four CARIBIC flight tracks are marked in cyan.





While there have been efforts to determine the relationship between Aerosol Index and aerosol optical depth and hence column loading, such relationships are frequently spatially variable owing to the sensitivity of the Aerosol Index to both the aerosol absorption optical depth and the altitude of the aerosol cloud (e.g. Christopher et al., 2008). BBA in the stratosphere will likely be associated with a stronger AI than a similar column loading of BBA in the troposphere. A more quantitative assessment of the aerosol column loading utilising the AI is beyond the scope of this paper.

### 4.4 BBA: vertical structure

Figure 7 shows the vertical structure of the simulated and observed BBA cloud along the CALIPSO tracks shown in fig. 6. The timings of the NAME simulations match the CALIPSO overpass times, and so is shifted in time by a few hours compared to that shown in fig. 6; the timing of the model data is shown in each panel in the left hand column. The comparison between the model data and the CALIPSO sounding on the 18 June is poor, with no features in the model matching the height or location of features in the measurements. In particular, the large feature between 16 km and 18 km in the model data is entirely absent from the CALIPSO sounding. This feature corresponds to the model particles released above 16 km, and as stated above, it seems likely that the aerosol cloud did not initially reach above this height. Also missing is the large area identified as elevated smoke in the VFM (black) in the boundary layer between $57^o$N and $62^o$N. Given the relatively small scale of the BBA cloud smoke on this day compared to the more developed aerosol cloud over the following days, the comparison is sensitive to small offsets in time and space between the observations and the model output. However, shifting the model data by a few degrees did not improve the comparison. By 19 June the comparison is more favourable, with some features in the model data corresponding in height and location with features in the CALIPSO sounding that have been classified as elevated smoke in the VFM. While the structure in the model data at around 9 km between 57oN and 60oN is not present in the VFM, there is some indication of it in the attenuated backscatter, and it may be the case that the classification scheme has failed due to the signal being attenuated by the optically thick structure above. From the 20 June onwards the comparison improves as the aerosol cloud disperses into a larger structure and continues to be shaped by the synoptic meteorology. By the 22 June the structures in the CALIPSO soundings identified as elevated smoke or sulphate/other (dark grey) are all present in the model data, and by the 23 and 24 June there is good agreement between the two in terms of structure and position.

Figure 8(a) shows the relative attenuated backscatter product from the Washington MPLNET lidar between 00:00 on the 22nd and 23:59 UTC on the 24 June. The panel (c) shows the NAME BBA concentration for the grid box over Washington DC for the same time. Although it is initially partly obscured by clouds, a well defined aerosol layer arrives over the site at around midday on the 22nd. Beginning as a single layer at an altitude of 5 km, the layer then rises slightly to 8 km, before descending to around 4 km and splitting into upper and lower layers, separated by around 500 m, by mid-morning on the 23 June. The layers then persist until midnight on the 23 June. The mean PDR with these layers (at 532 nm) is 8±2%, consistent with the layer being comprised of BBA (Groß et al., 2015a; Adam et al., 2020). Using a specific extinction value of 5.6 $m^2g^{-1}$ (calculated using Mie scattering theory and assuming a refractive index of 1.5 - 0.07i at 532 nm, a density of 1.9 g $cm^{-3}$, and a size distribution taken from (Petzold et al., 2007) to transform the extinction coefficient product to a mass concentration estimate, the mean backscatter weighted concentration within the layers is 15±5 $\mu gm^{-3}$.

**Figure 7.** Left hand column: Vertical structure of the NAME BBA mass concentrations along the CALIPSO tracks shown in fig. 6 [mgm$^{-3}$], middle column: CALIPSO Total Attenuated Backscatter at 532nm [m$^{-1}$sr$^{-1}$], and right hand column CALIPSO vertical feature mask. The NAME data is taken from the output file closest to the time of the CALIPSO overpass and so is shifted in time slightly compared to that shown in fig. 6. Key to VFM colour bar: 0 = not applicable, 1 = marine, 2 = dust, 3 = polluted continental / smoke, 4 = clean continental, 5 = polluted dust, 6 = elevated smoke, 7 = dusty marine, 8 = polar stratospheric cloud aerosol, 9 = volcanic ash, 10 = sulfate / other.





The general shape and position of this aerosol layer is also represented in the simulations, with good agreement for altitudes and timing. The size of the separated structure shown in the lidar data is on the order of the 500 m altitude grid on which the NAME data is reported, and it is therefore unsurprising that this has not been resolved by the model. By scaling the amount of mass released in the Alberta fires source term, it is possible to match the model concentrations above the Washington DC site to the lidar concentration estimates. Scaling the altitude bins uniformly from the ground to 16 km results in 0.2 Tg of particulate

material being released in the model.



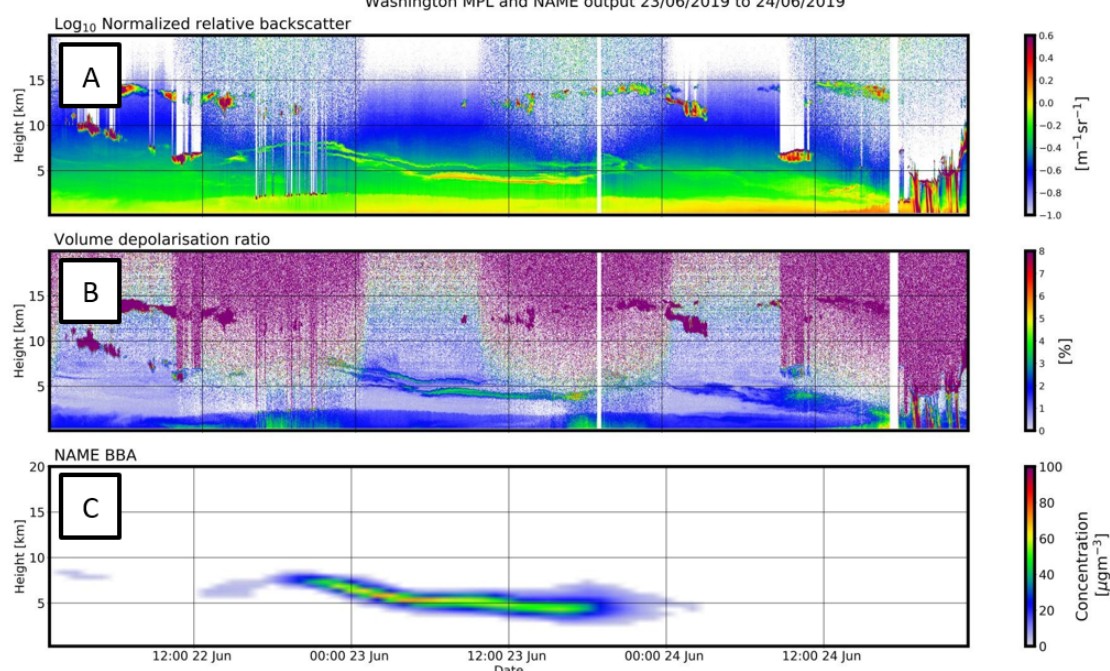

**Figure 8.** Washington MPL lidar data: a) normalized relative backscatter [m-1sr-1], b) Volume depolarisation ratio [%]. c) NAME air concentrations for BBA cloud [$\mu gm^{-3}$]. The NAME data has been scaled to match the lidar concentrations estimates. Data for 22 to 24 June 2019

## 4.5 BBA regional analysis over North America

Figure 9 shows the NAME simulated mass concentrations of BBA along the IAGOS-CARIBIC flight tracks marked in fig. 6. Overlaid on the model data are the flight paths colour coded to show the particle numbers (per cm$^3$ of air) counted by the Grimm OPC on board the IAGOS-CARIBIC aircraft. For flight F570 on the 19 June and during flight F571 on the 20 June there
is remarkable agreement between the locations in which the aircraft encountered elevated particle numbers, and the location of the simulated aerosol cloud. This is particularly true of flight F571, where the particle counts indicate where the aircraft has encountered an initial region of elevated particle counts at 10 km, centred on 62$^o$N, followed by a relatively particle free region, and then a second region of elevated particle counts at around 77$^o$N, all of which agree well with the aerosol cloud locations in the model. Flight F572 has not encountered any areas of elevated particle counts. According to the model, it has flown under
the large structure that extends out to pacific coast (see fig. 6); however, as stated above, this structure does not appear in the OMPS or CALIPSO measurements and is likely a model artefact. Flight F573 has flown between two separated parts of the model aerosol cloud. On this occasion, the locations of the CALIPSO track and the flight track cross at around 63$^o$N (although the flight track is between 06:00 and 07:00 and the CALIPSO overpass at 08:44), and a similar separated structure can be seen in the CALIPSO soundings on the 21 June at around 63$^o$N.



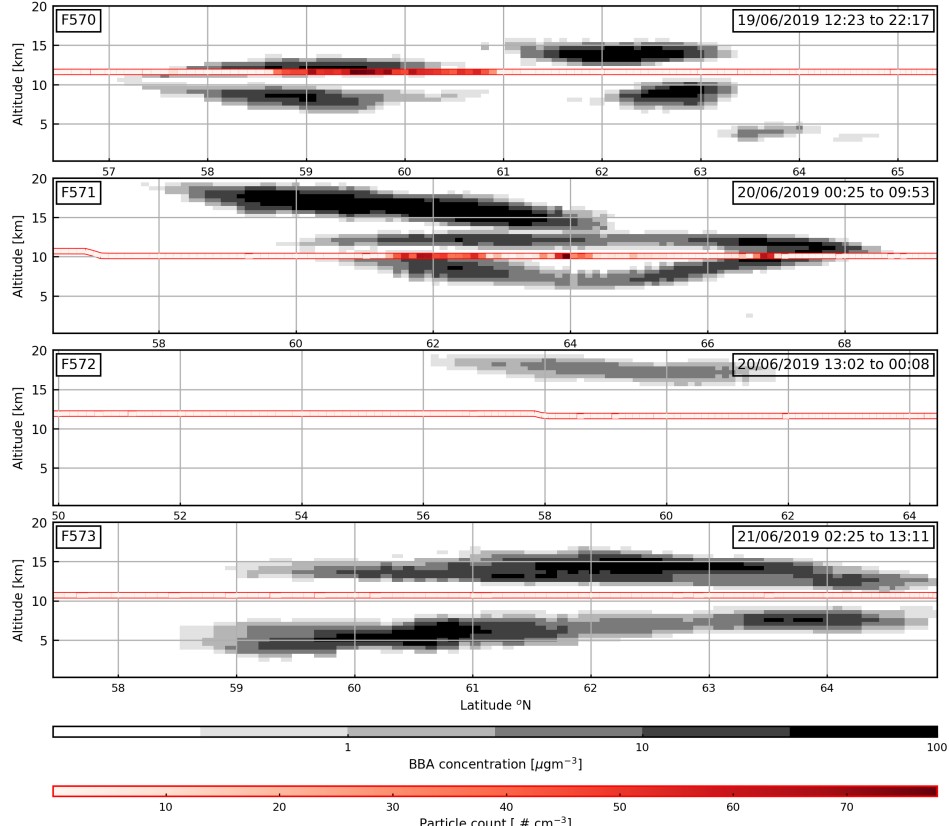

**Figure 9.** NAME air concentration data [$\mu$gm$^{-3}$] along the CARIBIC tracks shown in fig. 6. Overlaid are the CARIBIC flight paths coloured to show the particle count [number per cm3] recorded by the Grimm OPC. The NAME data is a mean of the one hour output files corresponding to the time the aircraft was transiting the aerosol cloud (times shown in panels).

So far we have presented the observation data and NAME simulations together, charting the dispersion of the volcanic and BBA clouds over the first five to six days. The observations indicate that the NAME simulations reflect the general pattern of the dispersion seen in the various observations, and as such can be used to gain some insight into the dispersion of the aerosol clouds in areas where there are no observations. The availability of observations as the aerosol clouds crossed the Atlantic towards the UK is sparse, as the concentrations were mostly below the detection limits of passive satellite observation, and so

we now rely upon only the simulations as the aerosol clouds move towards the UK and mainland Europe. Figure 10 shows the column loadings for all three simulated aerosols on the 27 June, the 1 July, and the 11 July. The Met Office operational dust forecast is also plotted with the BBA column loadings. The NAME simulations show the BBA cloud reaching the UK on the 27 of June, along with an aerosol cloud of Saharan dust. The Saharan dust moves away from the UK over the next four days, however the BBA cloud persists over the UK and is joined on the 1 July by the ash aerosol cloud and the first parts of the

sulfate aerosol cloud.

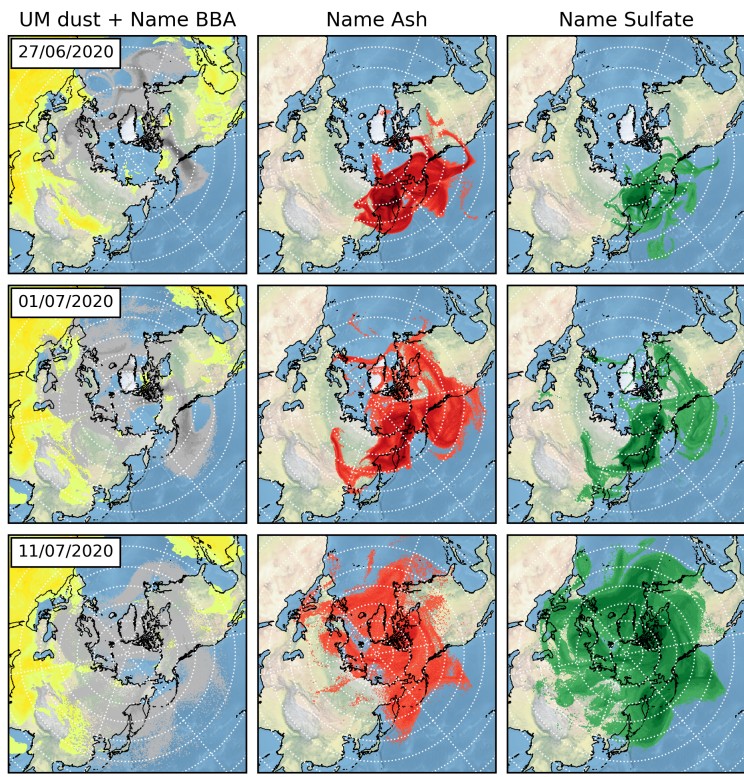

**Figure 10.** NAME dispersion model simulations for BBA (black/grey) (with UM dust field in yellow)), ash (red) and sulfate (green). Data for 00:00 UTC on date indicated in each panel.

Both simulated aerosol clouds continue to impact the UK over the next week to ten days, while also dispersing over much of the northern hemisphere. By 11 July the more substantial parts of the simulated sulfate cloud also reached the UK.

### 4.6 Regional analysis and Interpretation of material over the UK

Figure 11 shows examples of the data from the network lidars recorded on the dates displayed in fig. 10. The upper panels in each pair of axes show the log10 of the range corrected signal, and the lower panels show the log10 of the volume depolarisation ratio. Table 5 gives a summary of the aerosol layers measured across the network. To be consistent with the simulations the heights quoted are above sea level. As described below we have used long averaging windows to calculate backscatter and





PDR. However, even using this large averaging time it was not possible to measure the lidar ratio (which involves dividing a noisy signal by another noisy signal) as the results were so noisy as to be unusable.

The network first detected stratospheric layers over the Glasgow site on the evening of 27 June, with an upper layer at 14 km, and a lower layer at 11 km (axes A in fig. 11). There is also a much thicker layer in the troposphere between 2 km and 6 km after around 04:00 on the 28th. In this lower layer the mean PDR (at 355nm) was 26±3%. This is consistent with the layer being comprised of Saharan dust (Groß et al., 2013; Osborne et al., 2019), and the Met Office operational dust forecast confirms that this is the source of this lower altitude layer, and that it is unrelated to either the forest fires or the Raikoke

eruption. The measurements in the stratospheric layers are near the limit of the useful range of the lidars ($\approx$ 15km) and the layers were only detectable at night when the daylight background was absent. To calculate PDR values and mass estimates at this range we have used whole night averages (6 hours of continuous photon counting) to improve the signal to noise ratio as far as possible. We note that this assumes that the aerosol particle type and properties remain constant during that time. Using the whole night average, using the Raman and elastic channels, we calculate that the PDR in the upper and lower layers was

9±1% and 7±1% respectively. There are a very limited number of reported measurements of the PDR of stratospheric BBA at 355 nm. (Hu et al., 2018) report values of up to 26%, and (Haarig et al., 2018) report values around 20%. These values are considerably higher than our measurement. It is possible that the variation is due to differences in the production and ageing process of the aerosols in this layer compared to those measured in the previous studies. Tropospheric measurements of PDR of BBA are much more numerous, and are often reported to be below 5% (e.g. Groß et al., 2015b; Illingworth et al., 2015;

Adam et al., 2020), but the range in reported values is quite large, and (Burton et al., 2015) and (Vaughan et al., 2018) report tropospheric values of 17% and more. There is discussion in the literature as to the cause of the variation in the tropospheric measurements. Possible explanations being that some smoke layers contain irregularly shaped soil and dust particles lofted at the same time by the strong convection associated with wildfires, or alternatively that the aggregation of small soot particles as part of the ageing process can produce significantly non-spherical agglomerations (e.g. Jahl et al., 2021). It is possible that

a similar variation exists in the stratosphere. The concentration in the layers was small, with the backscatter weighted mean concentration in the upper layer being 14±8 $\mu$gm$^{-3}$.





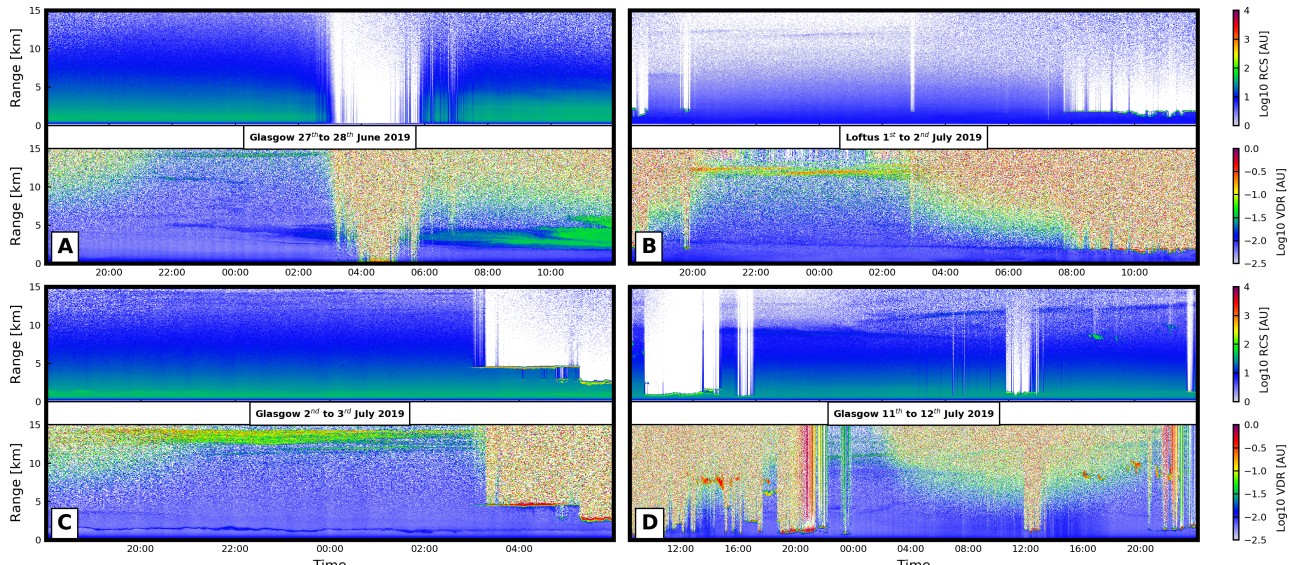

**Figure 11.** Lidar data from the Met Office lidar sites at Glasgow and Loftus on dates indicated in each axis pair A, B C and D. The upper panel of each pair shows the range corrected signal [AU], and the lower panel shows the volume depolarisation ratio (VDR) [%].

Rain and low cloud prevented any further measurements in the stratosphere until the evening of 1 July, when several UK network lidars detected aerosol layers between 11.5 km and 14 km. Axes B shows the data from Loftus for the 1 July and into the 2 July. Again using whole night averages, the mean PDR in these layers was 28±4% on 1 July into 2 July, and 32±7% on

2 July into 3 July (fig. 11(c)). This is a marked change from the PDR measured on the 27th/28th, and while it is not possible to unambiguously identify the aerosols using the PDR values alone, taken together with the arrival times of simulated BBA and ash clouds, we interpret this change to significantly higher PDR values as the arrival of the ash cloud, and the transition of the dominant aerosol type from BBA to volcanic ash. The maximum concentration was observed at Loftus and was 135±78 $\mu$gm$^{-3}$.

Low cloud and rain again prevented usable lidar measurements until 11 July when the lidars at Camborne and Glasgow detected an aerosol layer at 9.5 km, and an optically very thin layer (AOD ≈ 0.005) at 11 km (axis pair D in fig. 11). The PDR in the lower layer was 0.9±0.2%, and 13±5% in the higher layer. On 13 July the lidars detected a layer at 13 km with a mean PDR of 1.6±0.4%. Again, this is a marked change from the previous PDR measurements on 4 July, and when viewed together with the arrival times suggested by the NAME simulations, we interpret this as the arrival of the sulfate cloud, and that by 12

July the scattering in the stratospheric layers was dominated by spherical sulfate droplets.

## 5   Conclusions

The early arrival of aerosols layers over the UK in late June 2019 has been established to be due to the emissions of biomass burning aerosols from intense fires in Alberta that penetrated the tropopause on 17 June 2019. Volcanic ash and sulfur species





**Table 5.** Summary of lidar detections. There were marked changes in the PDR measured within the aerosol layers correlated to the dominant aerosol type

| Location | Time | Layer height [km] | PLDR [%] | Max concentration [$\mu gm^{-3}$] |
|---|---|---|---|---|
| | | 27 & 28 June 2019 | | |
| Glasgow | 21:30 to 03:30 | 10km to 11km | 7±0.5 | 9.2±6.3 |
| | | 14km to 14.5km | 9±1.0 | 13.3±8.3 |
| | | 1 to 2 July 2019 | | |
| Glasgow | 21:30 to 04:00 | 11.8 to 12.6km | 26±3.0 | 231±33 |
| Loftus | 21:30 to 04:00 | 11.5km to 12.5km | 28.8±5.2 | 148±76 |
| | 21:30 to 23:00 | 13km to 13.8km | 32.1±18 | 146±127 |
| Camborne | 21:30 to 04:00 | 13.5 to 13.9km | 24.5±4.4 | 21±9.5 |
| | | 3 to 4 July 2019 | | |
| Camborne | 00:05 to 04:00 | 14.1km to 14.7km | 30.2±7.5 | 38.6±25.9 |
| | 22:30 to 23:55 | 14.2km to 14.5km | 33.2±18.2 | 8.6±6.1 |
| Loftus | 01:15 to 02:40 | 13.2 to 13.8km | 32±10 | 135±77.7 |
| | | 12, 13 & 14 July 2019 | | |
| Glasgow | 01:30 to 03:00 (12) | 8.0km to 9.8km | 1.0±0.1 | 95.0±29.1 |
| | | 10.8km to 11.1km | 13.8±4.5 | 30.0±15 |
| Camborne | 22:00 to 03:30 (14) | 13.8 to 14.5km | 2.3±0.7 | 23.6±10.9 |
| Glasgow | 01:30 to 03:30 (14) | 13.2 to 14.6km | 1.1±0.4 | 14.5±3.8 |

injected into the upper troposphere and lower stratosphere by the 21 June 2019 eruption of Raikoke reached the UK around ten

days after emission, along with sulfate particles formed by the chemical transformation of volcanic $SO_2$. The UK lidars were able to make measurements of the particle linear depolarisation ratios and estimate the concentrations within these layers over the UK.

We have presented results from NAME model simulations alongside observations charting the evolution of the aerosol clouds produced by the Raikoke eruption and the Alberta forest fires. The results of the simulations were used to aid in the

interpretation of the UK lidar data. A mostly qualitative comparison between the observations and the simulation over the first five to six days after emission showed that the simulations were representative of the evolution of the aerosol clouds as resolved by the observation, and as such could give some insight into the dispersion of the aerosol clouds where no observations were available. This was particularly true of the simulations of the forest fire aerosol cloud. After around five to six days following emission, the aerosol clouds were no longer visible in the satellite observations. However, the dispersion simulations showed



that the BBA arrived over the UK by 27 June, the ash by 1 July and the main bulk of the sulfate aerosol cloud by 11 July and this was confirmed by the UK lidar network.

By scaling the total mass of BBA released in the simulations so that the air concentration in the distal aerosol cloud over Washington DC matches the concentration estimated from the Washington DC MPLNET lidar observations, we estimate that the Alberta fires released around 0.2 Tg of fine mode BBA. The mass of volcanic ash released in the simulations (estimated

using the Mastin relationship with plume top height) was not scaled to match observations. The simulated ash and sulfate concentrations above Fairbanks, Alaska (400 $\mu gm^{-3}$ and 25 $\mu gm^{-3}$ respectively) were on the same order of magnitude as concentrations estimated from the Fairbanks MPLNET lidar observations (250 $\mu gm^{-3}$ and 10 $\mu gm^{-3}$ for ash and sulfate respectively), suggesting that the mass of ash and $SO_2$ released in the simulations was reasonably representative of the real value .

The PDRs measured by the UK lidars were 1% in the sulfate-dominated aerosols, 9% in the BBA aerosol cloud, and 30% in the volcanic ash. These values did not allow an unambiguous identification of the material within the aerosol clouds, and it was not possible to measure lidar ratios within the layers over the UK. In this respect, NAME simulations were critical in interpreting the observations; they explained the change in the measured PDRs, from around 9% in the BBA cloud, over 30% in the ash-containing aerosol cloud, and around 1% in the sulfate dominated aerosol cloud. Importantly in terms of scientific

advice given to the London VAAC the dispersion simulations showed that the layers detected on the 27/28 June 2019 were in fact comprised of BBA particles, and not a mix of volcanic ash and sulfate particles. The low aerosol concentrations observed over the U.K during the period under study mean that the mistaken identification of the initial aerosol layers at the end of June would have negligible consequences. However, this study shows how the synergy of dispersion modelling and multiple observation sources is important in building a more complete description of aerosol loading.

*Author contributions.* M.O. and J.dL set up and conducted the NAME simulation with advice and guidance from C.W, F.B. and N.K.. N.K provided the set up configuration for the initial simulations. C.W, A.S., J.H and F.M. contributed to the scientific discussions and advised on data analysis. C.S. processed the Himawari 8 data and advised on interpretation. E.W and J.F. collected and processed the NASA-MPL lidar data. M.O and J.B. collected and process the U.K lidar data. A.P., U.B. and A.G. collected and processed the CARIBIC data. M.O. was responsible for general data analysis, manuscript preparation and plot preparation. FM and JH were PhD supervisors for MO and advised

and assisted with the data analysis and structuring of the article. All authors reviewed the manuscript.

*Acknowledgements.* Funding for PhD work for M.O. was provided by NERC through the University of Exeter, grant NE/M009416/1. Contributions from J.H. and A.S. benefited from support by the NERC ADVANCE (Aerosol-cloud-climate interactions deduced using Degassing VolcANiC Eruptions), grant NE/T006897/1. J.dL. & A.S acknowledge funding from the Natural Environment Research Council (NERC) V-PLUS grants NE/S00436X/1 and NE/S004025/1. We would like to thank all the Met Office teams involved in the VA lidar project. We

are grateful to the Civil Aviation Authority and Department for Transport for funding the lidar project. This work used JASMIN, the UK collaborative data analysis facility (doi:10.1109/BigData.2013.6691556), to run the NAME simulations.



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
