# Peer review of "The 2019 Raikoke volcanic eruption part 2: Particle phase dispersion and concurrent wildfire smoke emissions"

_Atmospheric Chemistry and Physics, 2021_

## Referee Comment (RC1)

The 2019 Raikoke volcanic eruption part 2: Particle phase dispersion and concurrent wildfire smoke emissions

Martin Osborne et al.

This paper presents a detailed study of the different kinds of aerosol present in the stratosphere over the Northern Hemisphere in late June and early July 2019. The use of the NAME model, together with satellite and ground-based observations for verification, allows the authors to disentangle volcanic ash, volcanic aerosol and biomass burning aerosol and therefore to determine the composition of the aerosol measured over the UK during this period. The paper is thorough and well written, and I have only minor comments on the manuscript.

Minor comments

p.4 l.88 comma can't be used to separate two full sentences.

p.7 l.156 Raikoke eruption

p.7 l. 171. What does 'unoptimised' mean? I found this section hard to follow – the ash profile is set equal to the $SO_2$ profile from IASI but the $SO_2$ profile is set to something different! I think the text would be easier to follow if the $SO_2$ and ash sections were swapped round and the text modified accordingly.

p.9 l.194. This sentence leads the reader to expect results from all ten lidars but in fact only Glasgow, Loftus and Camborne produced data used in this paper. That point should be made here.

p.13 l.261 The right-hand column shows sulfate aerosol, not the left-hand one.

p.13 l.266 Please point out the 'two distinct structures' more clearly. Also 'Figure 3 shows that, in the model…….'

p.13 l. 271 $SO_2$

p.14 l.280 'classified as'

p.14 l.285 'are too weak to be identified'

Fig 5 caption is not compatible with figure – these are lidar data not OMPS

p.14 l.308. Here a value of 0.01% is given for the sulfate pdr whereas on l 297 a value of 1% is quoted. They need to be consistent

p.15 l.325. Text is confusing. What is meant by 'higher layers'? The maximum ash concentration in fig 5b (10 km, afternoon of 26 June) is ~700 $\mu gm^{-3}$ not 400, and for sulfate (5c) it is 2 not 25 $\mu gm^{-3}$ . These values are not consistent with lines 310-314, and for sulfate

the concentration is lower than the 10 $\mu gm^{-3}$ estimated from the lidar.  This section needs re-writing to be consistent with the figure. The erroneous values are repeated in the Conclusions (line 491) which also needs revision.

p.17 l.337. Surely 'north-westerly jet' – the motion is towards the south-east

p.19 l.379. 59°N and 60°N

Fig 8 caption, $m^{-1}sr^{-1}$

p.22 l.408. 77°N is far beyond the range of fig. 9b – presumably this should be 67°
p.25 l.450 reference call-outs wrongly formatted

p.25 l.452 'Possible explanations are'

p.26 l.458 Panels B not Axes B

p.29 l. 532 article number (art. no. D00U02) should be given rather than n/a for page number

p.30 l.557 reference incomplete (needs name of book and its editor(s) as well as the Chapter)

p.30 l.560 reference needs putting in proper format

p.31 l.598 article number missing - also l.632, l.635, l.665, l.673, l.689, l.733, l.745, l.768

p.31 l.623 Journal shouldn't be capitalised

p.33 l.668 reference not correctly formatted

p.33 l.771 needs page numbers and doi

p.34 l.713 is this an article number?

p.35 l.747 No journal given, or doi

p.35 l.770 reference incorrectly formatted (two page number ranges, text before year is anomalous)

General typos
    -   the paper often uses the syntax 'on the 2 July' when the correct syntax is simply 'on 2 July' – please correct all such errors throughout the paper. (Sometimes the correct syntax has been used so the paper needs to be consistent)
    -   Each reference has two web links, which are often the same. Only one is needed – it is customary when there is a doi to give just that
    -   Where the paper references web pages, the date of last access is needed
    -   There are a lot of references to Discussion papers, some of them many years old. Either these should be replaced by the accepted journal article or they should be replaced by another reference – it should be assumed that papers that get stuck in Discussions have been rejected.
    -   The authors should check the references carefully as I'm sure I have missed many errors. This should not be left as a task for the copyeditors.

---

## Referee Comment (RC2)

[referee-annotated manuscript omitted]

---

## Referee Comment (RC3)

[referee-annotated manuscript omitted]

---

## Author Comment (AC1)

**Response to reviews of "The 2019 Raikoke volcanic eruption - Part 2: Particle phase dispersion and concurrent wildfire smoke emissions" by Osborne et al.**

We extend our thanks to the three reviewers for their appreciation of our work and the careful reviews and help in improving this paper. The paper is fairly long, and we are therefore particularly grateful to the reviewers for their time in reviewing the paper and their detailed responses.

We are glad that reviewers liked the paper and consider it worth publishing after addressing their points.

We hope we have addressed the points raised. Please find below our responses (in red) to the reviewers comments (in black).

**1 Response to reviewer 1.**

This paper presents a detailed study of the different kinds of aerosol present in the stratosphere over the Northern Hemisphere in late June and early July 2019. The use of the NAME model, together with satellite and ground-based observations for verification, allows the authors to disentangle volcanic ash, volcanic aerosol and biomass burning aerosol and therefore to determine the composition of the aerosol measured over the UK during this period. The paper is thorough and well written, and I have only minor comments on the manuscript.

Minor comments
p.4 l.88 comma can't be used to separate two full sentences.

Sorry - this is a typo – 'and' inserted after comma.

p.7 l.156 Raikoke eruption

Space inserted

p.7 l. 171. What does 'unoptimised' mean? I found this section hard to follow – the ash profile is set equal to the SO2 profile from IASI but the SO2 profile is set to something different! I think the text would be easier to follow if the SO2 and ash sections were swapped round and the text modified accordingly.

We have removed reference to an "un-optimised" profile, and swapped the sections around as suggested. The wording has been updated to reflect the new order, and we hope this section is now clearer.

p.9 l.194. This sentence leads the reader to expect results from all ten lidars but in fact only Glasgow, Loftus and Camborne produced data used in this paper. That point should be made here.

Agreed – the sentence "Data recorded by three of the network lidars, Camborne, Loftus and Glasgow, are presented here." has been added.

p.13 l.261 The right-hand column shows sulfate aerosol, not the left-hand one.

Corrected

p.13 l.266 Please point out the 'two distinct structures' more clearly. Also 'Figure 3 shows that, in the model…….'

"Corresponding to the red and yellow areas in the RGB image" added to better indicate which structure is being discussed.

"Figure 3 shows that the model, the aerosol clouds" deleted and replaced with "The simulated aerosol clouds"

p.13 l. 271 SO2

Corrected

p.14 l.280 'classified as'

Corrected

p.14 l.285 'are too weak to be identified'

Corrected

Fig 5 caption is not compatible with figure – these are lidar data not OMPS

Apologies – this is indeed the wrong figure caption. Now replaced with "Fairbanks MPL lidar: a) normalized relative backscatter [$m^{-1}sr^{-1}$], b) & c) NAME simulated air concentrations for ash and sulfate respectively [$ugm^{-3}$]. Data for 24 to 28 June 2019. The red outlines labelled 1, 2 and 3, and the vertical red lines are discussed in the text."

p.14 l.308. Here a value of 0.01% is given for the sulfate pdr whereas on l 297 a value of 1% is quoted. They need to be consistent

This is a typo – changed to 1%

p.15 l.325. Text is confusing. What is meant by 'higher layers'? The maximum ash concentration in fig 5b (10 km, afternoon of 26 June) is ~700 µgm-3 not 400, and for sulfate (5c) it is 2 not 25 µgm-3 . These values are not consistent with lines 310-314, and for sulfate the concentration is lower than the 10 µgm-3 estimated from the lidar. This section needs re-writing to be consistent with the figure. The erroneous values are repeated in the Conclusions (line 491) which also needs revision.

Apologies, the color-bar tick labels (manually placed to display actual concentration values rather than log10 values) were incorrectly placed in figure 5b and 5c, but the values quoted in the text are correct. The plot has now been corrected.

p.17 l.337. Surely 'north-westerly jet' – the motion is towards the south-east

Agreed – now corrected

p.19 l.379. 59°N and 60°N

Corrected

Fig 8 caption, m-1sr-1

Corrected

p.22 l.408. 77°N is far beyond the range of fig. 9b – presumably this should be 67°

Apologies – now changed to 67°

p.25 l.450 reference call-outs wrongly formatted

Brackets removed from references

p.25 l.452 'Possible explanations are'

Agreed – now changed

p.26 l.458 Panels B not Axes B

Now changed to "panels"

p.29 l. 532 article number (art. no. D00U02) should be given rather than n/a for page number

Corrected

p.30 l.557 reference incomplete (needs name of book and its editor(s) as well as the Chapter)

Apologies. Now corrected

p.30 l.560 reference needs putting in proper format

Corrected

p.31 l.598 article number missing - also l.632, l.635, l.665, l.673, l.689, l.733, l.745, l.768

Apologies – now corrected, we hope the article numbers are now displayed correctly.

p.31 l.623 Journal shouldn't be capitalised

Corrected

p.33 l.668 reference not correctly formatted

Corrected

p.33 l.771 needs page numbers and doi

Corrected

p.34 l.713 is this an article number?

This is a typo - corrected

p.35 l.747 No journal given, or doi

Corrected

p.35 l.770 reference incorrectly formatted (two page number ranges, text before year is anomalous)

Corrected

General typos
- the paper often uses the syntax 'on the 2 July' when the correct syntax is simply 'on 2 July' – please correct all such errors throughout the paper. (Sometimes the correct syntax has been used so the paper needs to be consistent)

This has now been corrected – we hope we have caught all the typos.

- Each reference has two web links, which are often the same. Only one is needed – it is customary when there is a doi to give just that

Agreed - URL removed where there is also a doi number.

- Where the paper references web pages, the date of last access is needed

Apologies – we hope the last access dates are now included

- There are a lot of references to Discussion papers, some of them many years old. Either these should be replaced by the accepted journal article or they should be replaced by another reference – it should be assumed that papers that get stuck in Discussions have been rejected.

Apologies – the final papers should have been referenced. We have updated the references. One paper did not have a final version, and has been removed.

- The authors should check the references carefully as I'm sure I have missed many errors. This should not be left as a task for the copyeditors.

We have looked through the references and hope we have caught the errors.

**2 Response to reviewer 2.**

This paper describes the appearance of wildfire smoke at the United Kingdom at around the same time as an ash cloud was expected from Raikoke volcano in the Kurile Islands. The distribution and composition of the clouds were analyzed from satellite and lidar measurements, and the Met. Office dispersion model NAME was used to compare the simulated and observed arrival times. The different clouds could not always be distinguished based on composition, e.g. from particle depolarization ratios in lidar, and so model simulations were required to distinguish them.

The study takes advantage of the rare opportunity to compare ash-cloud simulations with real data over Europe. And the presence of a smoke cloud adds a twist and makes the study unique and

highly worthy of publication.  The paper is well written, figures are clear and well explained, and the inferences seem to be well supported by the data.

I have made comments throughout the attached pdf.  None point to any significant flaws.

The most significant of the minor comments are below:

In Section 3.1.2, the source of plume height and height-mass distribution of ash, used to model ash transport at Raikoke, is said to come from a report at https://wiki.earthdata.nasa.gov/display/volres. I was unable to find a report at that site. L164 & L171 - Is this report supposed to be on this site? I can't find a report on that site.  Just a sort of bulletin board with posts about the Raikoke eruption.

Apologies. The report does not seem to be available  / published – wording charged to "as reported by the VolRes group"

In Figure 5, the caption doesn't seem to match the figure.

Apologies – now replaced with the correct caption

Line 312 and earlier: what particle density are you assuming when converting lidar backscatter data to particle concentrations?  How does uncertainty in this value affect uncertainty in concentration?

The specific extinction value of 0.6 $gm^{-2}$ assumed for volcanic ash was taken from Ansmann et al. 2011 and Marenco et al 2011, without assuming a density ourselves. These studies use a density of 2.6 $gcm^{-2}$ for volcanic ash. Uncertainty in this value will add greatly to the uncertainty in the specific extinction. In our mass concentration estimates, the total error in the specific extinction was dealt with by treating the 0.6 $gm^{-2}$ value as the mean of a normal distribution with a standard deviation of 10%.

The assumed densities for sulfate and BBA are listed in table 3. The calculated specific extinctions (using the particle size distributions listed in table 3)  for these aerosols were also treated as the mean of normal distributions with std = 10%

The Monte Carlo simulations then calculated the mass concentrations by randomly selecting values for the specific extinctions within these normal distributions (as well as similarly random values for other parameters and the lidar signals themselves, drawn from distributions based on literature values, or a Poisson distribution in the case of the lidar signals.

Wording added at line 222 to clarify.

Line 184: it's not clear how the stacked cylinders in Fig. 1b define the source mass release in NAME.  Does each cylinder release the same number of particles?  Do particles originate from random locations within each cylinder?

The mass was released uniformly throughout each cylinder on the same number of particles. Lines added here to clarify this, and an extra panel added to figure one.

I look forward to seeing the paper officially published.

Larry Mastin

Thank you for your positive comments. Below we have listed the comments added to the supplement PDF (here in black) and our responses (in red).

L65 – What is the UTLS? I don't see a definition. Upper Troposhere and Lower Stratosphere?

UTLS now defined as upper troposphere and lower stratosphere

L68 – It would be helpful to add a sentence explaining what depolarization ratios are characteristic for ash and other materials, like smoke.

Wording added to clarify:

While depolarisation ratios for volcanic ash are often found to be greater than 30%, and those for volcanic sulphate aerosols are expected to be around 1% (e.g. Illingworth at al. 2015), an external mixture of the two aerosols could produce an overall particle depolarisation ratio of 9% (Gross et al. 2011).

L76 – All clouds? even ash clouds?

Clarified as meaning meteorological clouds

L90 – for which five years?

Clarified as the previous 5 years, and wording added to state that it is the largest fire season since 1981.

114 – The observations were of poorer quality? Or the agreement between model and observations was of poorer quality? Why would the observations depend on the diffusion parameter?

We found that the lower quality observations ash meant that we were unable to observe if any improvement / better agreement between model and observations was achieved by altering the diffusion parameter. Wording added to clarify this.

127 – processing > changes

Corrected

155 – How close? A few hundred km?

We make the assumption that it is close enough to the volcano that it has no influence on the subsequent modelling and observations, but cannot know how close. Wording changed to:

"assumed to have fallen out close enough to the volcano so as to have no influence on the subsequent observations / simulations"

L163 Not a fraction in this case

Agreed - wording added to clarify

Figure caption 2 - Did this involve model inversion, like that of Eckhardt et al. (2008)? www.atmos-chem-phys.net/8/3881/2008/

The type of inverse modelling techniques described in Eckhardr were not used to produce this this profile – the process used is fully described in the appendix of the part 1 paper. Wording added here to clarify:

"following a detailed comparison between the SO$_2$ product from TROPOMI and NAME simulations (and as described in more detail on Appendix A of de Leeuw et al. 2021)"

L211 – E > W?

Apologies – corrected throughout!

L213 – are

Corrected

L222 - This technique is not explained. What are you modeling using a Monte Carlo technique? What inputs are being randomly selected? Is there a reference that describes these?

Wording here to clarify:

"Errors in both the lidar extinction and backscatter retrievals, and the calculation of Kext contribute to the uncertainties in the final mass estimates. Errors in the lidar retrievals have been calculated using a Monte Carlo technique (D'Amico et al. 2016), taking into account both the statistical errors in the raw lidar signals, and errors in the systematic parameters used in the processing, such as assumed lidar ratios and polarisation factors. The uncertainty on calculated Kext values has been estimated by propagating the error in the assumed aerosol densities. In total, the error in the mass estimates is on the order of +/-50%.

L280 - You mean that little blip around 49 degrees latitude in the Calipso image? Perhaps mention the latitude. I think a lot of people will miss it.

Agreed - Wording updated to include reference to latitude of 49 degrees N

L289 at > as

Corrected

L285 as not being > that they are not

Corrected

L294 I can't distinguish the light gray ash from the gray background.

Agreed – The VFM color scheme was chosen to reflect those on the NASA website, but does use three different grays. It is difficult to find ten discrete colors that are different enough to prevent

some clashes. We have changed the background to light pink in all vertical feature mask plots, we hope this makes the various aerosol types clearer.

L312 - What about uncertainty in particle density.

The uncertainty in particle density is captured as part of the uncertainties in specific extinction. We hope that the clarification at line 222 makes this clearer.

Figure 4 caption: is > are.

Corrected

Figure 4 caption: I can't distinguish this gray color from the background gray in the images

Background changed to light pink in all vertical feature mask plots

L327 – "allowed to remain in the distal field" - What does this mean?

Apologies, we mean the fine mass fraction of the total ejected mass allowed to remain in the simulations. Wording changed to

"the fine mass fraction of ash allowed to remain in the simulations"

L326 - Yes, I suppose. It seems like you're waving away a fairly significant discrepancy between the lidar data and the model results--the big blob in the NAME Ash plot on the afternoon of June 26. But it may take a lot of work to find the source of the discrepancy. I can understand your lack of enthusiasm in pursuing it.

The large structures in the simulation are a significant discrepancy! We tried to change the source term (running simulations releasing mass only at each 500m altitude increment independently) but this was not enough to remove this structure, which appeared to contain contributions from all release heights in the source term. A thorough investigation of the cause of this discrepancy is beyond our scope here. We have added wording at line 322 stating:

"Attempts at modifying the vertical structure of the ash source term by changing the mass released at each altitude could not remove this structure. A thorough investigation of the cause of this discrepancy is beyond the scope of this study, but we note that this is a significant difference between the simulations and lidar observations."

Table 4 Grid boxes in simulated aerosol cloud - Can this be stated as cloud area? What is the size of a grid box? Are there any figures that show the grid boxes?

Each simulated grid box is $0.2°$ latitude $\times 0.4°$ longitude at the surface - approximately 10km x 10km at mid latitudes. This too small to represent easily on the large-scale figures. This information has been added to the table header

L379 – degree symbol not superscripted

Corrected

L391 comprised > composed

Agreed – now corrected

L409 - Seems like the pilots would have noticed this.

There is no report of this from the pilots during this flight that we are aware of, but we do not have access to the pilots notes as this is a commercial airliner rather than a research aircraft. Our experience is that it is often the case that an aerosol plume is apparent to the pilots eye when viewed horizontally, but often not when viewed from above or below.

Figure 9 caption cm3 > cm3

Corrected

L422 - This can't really be seen in Fig. 10. It looks like there's a smoke cloud encircling most of the globe at the latitude of the U.K. in Fig. 10 on 27 June.

This is clearer in an animation - wording changed to:

"The NAME simulations show the BBA clouds having reached the UK by 27 June"

**3 Response to reviewer 3.**

This paper presents a study combining NAME model simulations with satellite and ground-based observations to determine the composition and origin of aerosol plumes over the United Kingdom. This involved analyzing the long-range transport of stratospheric aerosol in the Northern Hemisphere and the optical properties of the aerosol to identify the source of aerosol from biomass burning by wildfires in Canada, as well as volcanic ash and volcanic sulfate aerosol from the Raikoke volcanic eruption in June 2019. Finally, both signals could be well separated from each other using the described technique.

The paper is well written and I have mostly minor comments on the figures and technical corrections. I have made some technical notes in the attached pdf file.

**General comments:**

I have a question to clarify the implementation of emissions in your model simulations: You used "a series of stacked and staggered cylinders" to implement the vertical emissions from the forest fire. If I understand you correctly, you are using a column above the volcano with a vertical $SO_2$ emission profile from the first paper, but not individual cylinders as for the forest fire emissions? You also used a vertical profile for volcanic ash implementation.

Yes this is correct. The off-set cylinders were used to better represent the hot, buoyant smoke being advected / dispersed horizontally as it rises, as against the volcanic emissions that are better repented as an immediate upwards explosive ejection, which are then horizontally advected.

But I can't find a vertical profile for the implemented forest fires. So how did you scale the 0.1 Tg of material released in each cylinder? Could you briefly describe how you calculated the mass, size, height and position of these cylinders? Could you also provide a vertical profile for the implementation of the forest fires, similar to Figure 2?

We have added a new panel to figure one with a source profile. Each cylinder contained the same arbitrary mass of 0.1Tg released over 100k model particles. The size and height of the cylinders were based only on the MODIS image in panel A of figure one, and the expectation that the smoke had pierced the tropopause. Wording modified at line 182:

"To represent this in NAME we have used a series of stacked and offset cylinders. A conceptual diagram of this structure is shown in fig. 1b. An arbitrary mass of 0.1 Tg of material was released uniformly throughout each of the cylinders, on 100k number of particles, and the results later scaled to match observations (see section 4). Panel c of fig. 1 shows the concentration of model particles within each cylinder. The size and position of the cylinders was based only on the MODIS image shown in panel a of fig. 1."

 **Specific comments:**

p.15 l. 325: The value 25 μgm$^{-3}$ does not fit the scale in fig. 5c.

Apologies, as described above in response to reviewer 1's comnet - the colour-bar tick labels were manually placed to display actual concentration values rather than log10 values. These were incorrectly placed in figure 5b and 5c, but the values quoted in the text are correct. The plot has now been corrected.

Figure 4 and Figure 7: For some people, it might be difficult to distinguish between these colors: 0; 7; 9; 10

Agreed. The background gray has been hanged to light pink which we hope makes the other colors more apparent and easier to distinguish.

Figure 5: The caption does not belong to figure 5 -> figure 6.

Apologies – caption now replaced with the correct one

Figure 8b: Panel 8b is not described in the text. What is panel 8b (volume depolarisation ratio) needed for?

Panle 8b is the depolarisation ratio – now referred to in the text at lines 386 and 390 with wording as follows:

L386 - Panel 8b shows the volume depolarisation ratio.

L390 - As can be seen from panel 8b the layers are composed of depolarising particles. The mean PDR with these layers (at 532 nm) is 8±2%, consistent with the layer being comprised of BBA (Groß et al., 2015a; Adam et al., 2020).

Figure 11: The units in figure 11 are log10 VDR [AU] and log10 RCS [AU], but in the text you use PDR [%]. Maybe you might highlight the most important spots, as in Figure 5, to clarify the description in the text on p. 24-26.

Agreed – depolarising layers now outlined in red.

 **Technical corrections:**

You mix different spellings of sulphate/sulfate in the text e.g. p. 7 l.176/177. In part 1 paper you used sulfate.

Apologies – now corrected

End all captions of figures and tables with: ".".

Now corrected

Check that there is always a blank between the value and the unit.

Title: part 2: In the first part of your study you wrote: ... **-** **P**art 1: ...

Apologies – now corrected to match the format of the Part 1 paper

p.6 Table 1: double parenthesis ((1999); "blank" between (2007), used

Now corrected

p.7 l.156: "blank" between Raikoke eruption

Corrected

p.7 l.172: No reference for de Leeuw et al. (2021). Do you mean 2020?

Apologies – 2021 is the (now published) final version. Reference now corrected

p.8 Figure 2 l.4: the $SO_2$ product

Now corrected

p.9 Table2: at 532 nm

Corrected – all tables now use "at" in place of @

p.12 l.252: $cm^3$

Now corrected

p.13 l.261: right-hand column

Corrected

p.13 l.271: $SO_2$

Corrected

p.13 l.278: "blank" 532 nm

Corrected

p.14 l.320: "blank" 13 km

Corrected

p.19 l.379: 57°N and 60°N

Corrected

p.19 l. 393: Petzold et al., (2007)

Corrected

p.20 figure 7: "blank" 532 nm

Corrected

p.22 figure 8: [m$^{-1}$sr$^{-1}$]

Corrected

p.22 l.408: 77°N are out of range of figure 6.

Apologies – now corrected to 60oN

p.23 figure 9: [number per cm$^3$]

Corrected

p.25 l.436: panel A

Corrected

p.25: Check the parenthesis of your citations.

Apologies – parenthesis removed where appropriate

p.27 table 5: Formatting: X km for each entry

Corrected

References: Please double check the format of your references ONE BY ONE to make sure they are in the ACP reference format: e.g. n/a–n/a, two web links, …

We have reviewed the references and hope we have corrected the errors.